

# Statistical patterns of theory uncertainties

Aishik Ghosh[1,2], Benjamin Nachman[1,3], Tilman Plehn[4], Lily Shire[2],
Tim M. P. Tait[2] and Daniel Whiteson[2]

**1** Physics Division, Lawrence Berkeley National Laboratory, Berkeley, USA
**2** Department of Physics & Astronomy, University of California, Irvine, USA
**3** Berkeley Institute for Data Science, University of California, Berkeley, USA
**4** Institut für Theoretische Physik, Universität Heidelberg, Germany

## Abstract

A comprehensive uncertainty estimation is vital for the precision program of the LHC. While experimental uncertainties are often described by stochastic processes and well-defined nuisance parameters, theoretical uncertainties lack such a description. We study uncertainty estimates for cross-section predictions based on scale variations across a large set of processes. We find patterns similar to a stochastic origin, with accurate uncertainties for processes mediated by the strong force, but a systematic underestimate for electroweak processes. We propose an improved scheme, based on the scale variation of reference processes, which reduces outliers in the mapping from leading order to next-to-leading-order in perturbation theory.

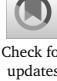

# 1 Introduction

A core goal of particle physics is to measure the parameters of the Standard Model (SM) and its extensions. For physics at the Large Hadron Collider (LHC), the SM is connected to observables via a combination of predictions for perturbative cross-sections, event generation, and detector simulations. To measure fundamental parameters, particle physics relies on likelihood functions extracted from experimental data and expressed as functions of the model parameters for a fixed dataset. With the likelihood functions, we can determine confidence intervals for the target. In addition to parameters of interest, such as *e.g.* the mass of the Higgs boson or a Wilson coefficient in an effective field theory, models depend on *nuisance parameters*, which describe systematic uncertainties, such as the modeling of the experimental response or auxiliary components of the theoretical predictions. The dependence on nuisance parameters is typically treated by profiling or marginalization [1–5].

With large datasets at next-generation facilities such as the high-luminosity LHC and the Deep Underground Neutrino Experiment (DUNE), systematic uncertainties will be a limiting factor for many important measurements. Therefore, a careful assessment of the statistical treatment of systematic uncertainties is critical. Despite a unified statistical treatment in practice, in reality, nuisance parameters describe sources of uncertainty which are quite disparate in their origin and statistical behavior. In one category are uncertainties due to auxiliary measurements which have inherent statistical uncertainty because of finite sample sizes. An example is the calibrated detector response from dedicated analyses, which is used to model the expected data under various hypotheses. This uncertainty would vanish for an infinite-sized calibration sample, and hypothetical similar finite calibration samples would be expected to yield different, typically Poisson-distributed, results due to the stochastic nature of the data.

Another category are uncertainties that arise due to our inability to perform infinitely precise calculations [6], or due to a lack of first principle theory predictions. These uncertainties are not determined by stochastic processes, in that hypothetical similar efforts would result in identical results if one were to repeat them making the same assumptions and approximations. As such, one does not probe a well-defined abstract space of theoretical possibilities when comparing different approaches based on different assumptions. In some cases, as in choice between two *ad hoc* models, such a distribution has limited ability to probabilistically encapsulate the space of possibilities. In other cases, such as uncertainties due to limited-order calculations of cross-sections, the interpretation of such distributions is at best unclear. The common practice of treating all of the nuisance parameters on the same statistical footing as if every case is drawn via a stochastic process from a well-defined distribution can result in misleadingly small estimations of the uncertainties in the derived parameters [4].

We explore this second type of uncertainty, focusing on fixed-order perturbation theory for inclusive cross-section calculations at the LHC. Our discussion begins in Sec. 2 with a review of the standard method of estimating the theoretical uncertainty due to limited-order calculations of inclusive LHC cross-sections via scale variations. In Sec. 3, we assess this method from a global perspective, examining inclusive cross-sections at next-to-leading order (NLO) and leading order (LO) in QCD for many available reactions. We examine the distribution of those predictions, revealing some surprising trends and behavior. We identify a class of electroweak processes for which leading-order uncertainty estimates fail spectacularly, and propose an alternative scheme with significantly improved performance in Sec. 4. Sec. 5 contains our outlook and conclusions.

## 2  Theory uncertainties from scale variations

The description of proton-proton collisions in the collinear approximation is based on the separation of hard scattering at the parton level convolved with universal parton densities encapsulating non-perturbative inputs [7, 8],

$$\sigma(\theta) \approx \sum_{a,b} \int dx_a x_b f_a(x_a; \mu_F) f_b(x_b; \mu_F) \hat{\sigma}_{ab}(\theta; \mu_F, \mu_R), \tag{1}$$

where the sum runs over partons inside the protons, $f_{a/b}$ are parton distribution functions (PDFs), $\hat{\sigma}$ is the partonic hard-scattering cross-section, and $\theta$ represents parameter(s) of interest. A typical LHC analysis infers information on the parameter of interest $\theta$ through comparison with data measuring the observable $\sigma(\theta)$. We will be concerned primarily with inclusive cross sections, which are under better theoretical control than differential cross-sections, including QCD radiation of jets, or fully exclusive event generation.

The partonic cross-section $\hat{\sigma}$ is computed as a perturbative expansion in the QCD coupling $\alpha_s$ [6], but estimating the precision of a given truncation is challenging. Estimates based on the size of the expansion parameter $\alpha_s$, or $\alpha_s/\pi$, are fraught because the perturbative series formally has a zero radius of convergence, implying that at a high enough order one does not expect subsequent terms in the expansion to be smaller than the preceding ones. While most predictions for inclusive cross-sections at the LHC appear to be convergent at least up to next-to-next-to-leading order (NNLO), naive estimates based on the size of the expansion parameter are found to be insufficiently conservative [6].

The PDFs appearing in Eq.(1) are extracted from a large orthogonal dataset (keeping track of the scale dependence) under the assumption that the relevant processes are described by the SM [9], and include their own comprehensive uncertainty treatment [10–13].

At each perturbative order, ultraviolet (UV) divergences in cross-section predictions are removed through renormalization, introducing a logarithmic dependence on an unphysical renormalization scale $\mu_R$ in the prediction. Similarly, infrared (IR) and collinear divergences are absorbed into the definition of the parton densities, introducing logarithms of an equally unphysical factorization scale $\mu_F$. Both scales can be related through the resummation of large collinear logarithms, but generally are independent scales with different infrared and ultraviolet origins and can be chosen independently [7].

The dependence of the theoretical prediction on the choice of scale introduces an arbitrariness into the prediction, and is an artifact arising from the truncation of the perturbative series. Formally, at all orders, the scale dependence would vanish, and thus the degree to which the prediction depends on the choice of scale at any finite order can be thought of as a measure of how significant the contribution of the uncomputed remaining terms in the series is expected to be. Traditionally, the uncertainty on the predicted hadronic cross-section in Eq.(1) is estimated by varying $\mu_R$ and $\mu_F$ around a suitable central scale. The choice of the central scale can vary depending on the physics process, it could be for example the scalar sum of transverse mass of all final state particles,the invariant mass of the system being produced, the average transverse energy of jets produced, or centre-of-mass energy of the collider. The logarithmic dependence on the scale requires $\mu \sim E$ where $E$ characterizes the physical energy scale appearing in the observable of interest. This choice insures that terms growing as $\log E/\mu$ do not spoil perturbation theory, but it cannot distinguish choices of scale differing by order-one factors.

While the scale variation may provide a measure of the impact of missing higher orders, it is not the 'true' uncertainty – which would quantify the likelihood distribution of the difference between the estimated cross-section and the all-orders prediction. Nonetheless, despite

the fact that the theory uncertainty on the cross-section prediction has no rigorous statistical interpretation, in fits to data, the corresponding nuisance parameters are often treated as Gaussian-distributed random variables. Trying to be reasonably conservative, one could instead think of a rate prediction as a range of expected values [1–5]

$$\sigma \in [\sigma_-, \sigma_+] \,. \tag{2}$$

The implementation of an allowed range in terms of a nuisance parameter is not trivial. A flat distribution of the corresponding nuisance parameter is not invariant under transformations such as changing the prediction from $\sigma$ to $\log \sigma$. It induces volume effects in the marginalization, which one can only partially avoid by using a profile likelihood.

Based on Eq.(2), a lower bound on the uncertainty can be estimated through the dependence on the unphysical scales at a given order in perturbation theory. For pure QCD processes, the dependence on the renormalization scale typically dominates, and the range of predictions effectively corresponds to a range of $\mu_R$,

$$\sigma \in [\sigma_-, \sigma_+] \equiv \left[ \sigma(\mu_{R,+}), \sigma(\mu_{R,-}) \right] \,, \tag{3}$$

and a suitable central scale choice $\sigma_0 = \sigma(\mu_{R,0})$ is often the transverse mass of the final state,

$$\mu_{R,0} = \sum_i \sqrt{m_i^2 + p_{T,i}^2} \,, \tag{4}$$

where the sum is over final state particles of mass $m_i$ and transverse momentum $p_{T,i}$. Assuming that the leading dependence on the renormalization scale enters through the running of the strong coupling, the scale-based uncertainty can be expressed

$$\Delta\sigma_{\text{scale}} = \frac{\partial \sigma}{\partial \alpha_s} \Delta\alpha_s = \frac{\partial \sigma}{\partial \alpha_s} \frac{\alpha_s(\mu_{R,-}) - \alpha_s(\mu_{R,+})}{2} \,, \tag{5}$$

where the appropriate range of scales defined by $\mu_{R,\pm}$ is discussed below. An approach based on lower and upper limits defined by a scale variation ensures that:

1. no $\mathcal{O}(1)$ variation of the unphysical scale around the central choice is considered more likely than any other;

2. there is no long tail of exponentially suppressed probability to obtain a very large deviation from perturbative QCD; and

3. perturbative and process-dependent arguments always leave an order-one uncertainty on the scale choice.

Two features of using scale dependence as the uncertainty measure are that it decreases as one considers higher order calculations, and that it increases when additional particles are added to the final state. We illustrate this for an $n$-particle production process at leading order in QCD and assuming that the renormalization scale only occurs implicitly through $\alpha_s$,

$$\sigma \propto \alpha_s^n \quad \Rightarrow \quad \frac{\Delta\sigma_{\text{scale}}}{\sigma_0} = n \times \frac{\alpha_s(\mu_{R,-}) - \alpha_s(\mu_{R,+})}{2\alpha_s(\mu_{R,0})} \,. \tag{6}$$

Until now we have not discussed the actual size of the scale variation. One can gain insight into the appropriate range of scales that one should consider in forming the uncertainty estimate from the process that is arguably the best-understood QCD reaction at hadron colliders, $t\bar{t}$

production. Experience from its perturbative behavior at the Tevatron [14–17] and at the LHC [18–22] motivates the standard choice:

$$\mu_{R,0} = m_t, \qquad \mu_{R,+} = 2m_t, \qquad \mu_{R,-} = \frac{m_t}{2}, \qquad (7)$$

such that the LO and NLO uncertainty estimates cover the known higher-order corrections. This recipe is simply generalized to QCD production processes of pairs of other heavy particles at hadron colliders. The LO scale dependence through the strong coupling is

$$\alpha_s(p^2) = \frac{1}{b_0 \log \dfrac{p^2}{\Lambda_{\text{QCD}}^2}}, \qquad \text{with} \quad b_0 = \frac{1}{4\pi}\left(\frac{11}{3}N_c - \frac{2}{3}n_f\right)$$

$$\Rightarrow \quad \frac{\alpha_s(p^2/4)}{\alpha_s(p^2)} \approx 1 + b_0\alpha_s(p^2)\log 4 \approx 1.04,$$

$$\frac{\alpha_s(4p^2)}{\alpha_s(p^2)} \approx 1 - b_0\alpha_s(p^2)\log 4 \approx 0.96. \qquad (8)$$

Around $\mu_R = m_t$, $\alpha_s^{(\text{LO})} \in [0.112, 0.120]$, and similar ranges are obtained at NLO and NNLO. Including higher orders in perturbation theory, the scale dependence of the $t\bar{t}$ production rate decreases through explicit appearance of $\log \mu_R$ in the rate prediction, which partially compensates the implicit dependence through $\alpha_s$. This cancellation is sensitively dependent on the details of the process, preventing one from obtaining a general rule to infer higher-order corrections (*K*-factors) for a wider selection of processes.

The same method with the same range of scales has been demonstrated to work for multi-jet production [23–29]. However, it fails *e.g.* in the case of Higgs production, where notably the NLO *K*–factor is much larger than expected [30–34]. In such cases one can often identify the reason for the larger-than-expected higher-order contributions, such as a loop-induced $(2 \to 1)$-scattering at LO, higher order contributions from new partonic channels, and/or large logarithms. For processes which are governed by more than one numerically relevant scale, for example the $\mu_R$ and $\mu_F$, one estimates the allowed range of the cross-section by using the extremal values in Eq.(3),

$$\sigma_- = \min_{\mu_R,\mu_F} \sigma(\mu_R,\mu_F), \qquad \text{and} \qquad \sigma_+ = \max_{\mu_R,\mu_F} \sigma(\mu_R,\mu_F). \qquad (9)$$

An alternative to including the full multi-scale dependence is to exclude the largest scale ratios [35].

Bayesian approaches [36] or mathematical approaches to perturbative series [37] attempt to parse the confidence interval on a particular inclusive rate in a different way, for processes in which higher order QCD corrections are known. The Bayesian approach estimates the size of the higher order corrections based on the inferred patterns based on the available terms, under the assumption that the perturbative series should converge upon a stable result at any fixed order (which is true in practice for known LHC predictions). By including process-specific information [38], they are able to infer a non-trivial shape for the distribution of the nuisance parameter, related to missing higher-order corrections from known higher orders in perturbation theory.

Such analyses typically settle on a very similar interval for the prediction as the naive scale-based method described above, though it remains difficult to meaningfully predict an optimal scale choice, even at the process-by-process level [35]. Ref [39] finds that they generally work well provided one corrects for generic features from QFT such as the factorial growth of the perturbative coefficients and expands the estimated size of the expansion parameter

from $\alpha_s \to \alpha_s/\lambda$ with $\lambda \simeq 0.6$ for hadronic processes. It further discerned that the interval corresponding to scale variation by a factor of two is not always sufficiently conservative at the 95% confidence level and reported an increased uncertainty for Higgs production, $\sigma(pp \to H)$, where, as mentioned above, specific features are known to spoil the scale variation estimate.

Consequently, while the flat distribution of Eq.(2) with range determined by scale variation by a factor of two, as suggested by $t\bar{t}$ production, does not encapsulate a nuanced shape for the corresponding nuisance parameter distribution, it remains a reasonably well-justified approximation that holds for a variety of hard processes. As we consider trends for a plethora of different processes, we adopt it below.

## 3  NLO dataset and results

To assess the standard scale variation method of assigning a theory uncertainty to predictions of inclusive cross-sections at the LHC, we analyze the comprehensive set of LO and NLO calculations presented in Ref. [40]. For each case, we investigate to what degree the uncertainty derived from scale variation of the LO prediction is a reasonable estimate of the difference between it and the known NLO calculation. While in many cases these NLO predictions are not the state of the art for the particular process at hand, and it would be very interesting to perform the same study using higher-order calculations, the number of NNLO calculations is comparably much smaller, which would limit the scope of the available processes. Furthermore, most searches at the LHC still use LO for generating signal samples, particularly for signal samples in supersymmetry and exotics searches and the computational cost of generating large NLO samples can be prohibitive also for other BSM searches.

We extract the central value and the scale uncertainties for 136 processes,[1] and neglect the statistical uncertainties from Monte Carlo integration, which are typically small in comparison, and PDF uncertainties, which are due to measurements in auxiliary datasets and so are in a different category and with complicated correlations between processes. All cross-sections are calculated by MADGRAPH5_AMC@NLO [41]. The Higgs boson is assumed to have $m_H = 125$ GeV, and the top quark $m_t = 173.2$ GeV. The parton distribution functions MSTWnlo2008 [42] are used with $n_f = 4, 5$. The choice of a central scale is

$$\mu_0 = \frac{H_T}{2} = \frac{1}{2} \sum_{\text{final state}} \sqrt{p_T^2 + m^2} \,. \tag{10}$$

The LO uncertainty is estimated by varying the factorization and renormalization scales by factors of two relative to the central scale, as defined in Eq.(9).

Given an estimator $x$ for the parameter $\mu$ with an estimated uncertainty $\Delta x$, we can analyze that uncertainty estimate using the pull

$$t = \frac{x - \mu}{\Delta x} \,. \tag{11}$$

If $x$ is a Gaussian random variable around $\mu$, then the pull will follow a Gaussian distribution with zero mean and unit variance. We treat $\sigma_{\text{LO}}$ as an estimator for $\sigma_{\text{NLO}}$ and analyze the behavior of its uncertainty $\Delta\sigma_{\text{LO scale}}$ by examining

$$t_{\text{scale}} = \frac{\sigma_{\text{NLO}} - \sigma_{\text{LO}}}{\Delta\sigma_{\text{LO scale}}} \,, \tag{12}$$

---

[1]All $pp$-initiated processes other than $pp \to \gamma W^+ W^- j$, for which the NLO cross-section has a typo, confirmed by the authors of Ref. [40] via private communication.

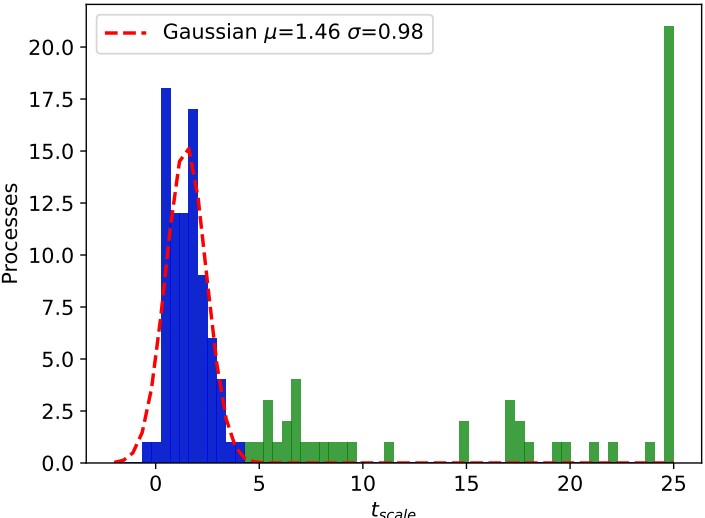

Figure 1: Performance of the uncertainty estimation in LO cross section calculations. Shown is the scale-based pull $t_{\text{scale}}$, defined in Eq.(12). Pulls greater than 25 are shown at 25. Blue entries with $|t_{\text{scale}}| < 4$ are included in the Gaussian fit (red).

using the positive $\Delta\sigma_{\text{LO scale}}$ if $\sigma_{\text{NLO}} > \sigma_{\text{LO}}$, and the negative otherwise. Note that in Eq.(12) we reorder the terms in the numerator relative to Eq.(11) in anticipation of $\sigma_{\text{NLO}}$ being larger than $\sigma_{\text{LO}}$; the statistical behavior is unchanged.

Figure 1 shows the distribution of the pull for our dataset. The distribution is clearly not Gaussian with zero mean and unit variance, though it displays some expected behavior. For example, the pull is almost always greater than zero, which aligns with our expectation that cross-section estimates tend to grow as additional partonic channels are included beyond LO. There also appears to be a core distribution which is locally approximately normally distributed with unit variance, indicating that for many processes, the central scale and the scale variation are accurate indicators for the NLO result.

In addition, there is a very long positive tail, indicating many processes, where the uncertainty is dramatically underestimated. The complete list of processes and the associated pulls are given in Tables 2-4. Many of the processes with underestimated uncertainties are those with many particles and without QCD vertices. Indeed, unlike pure QCD processes, QCD corrections to electroweak processes do not appear to be covered by the scale-based uncertainty estimate. Numerically, the leading scale dependence in QCD processes is the renormalization scale, and this scale dependence is absent in electroweak processes at LO, and is small at NLO. For instance, di-lepton production at the LHC is a purely electroweak process and only has a small factorization scale dependence at leading order, which does not cover the NLO corrections. This problem extends beyond Drell-Yan [43], to di-boson [44], and tri-boson production [45].

In addition to, generally predicting uncertainty estimates which are too small, electroweak processes encounter large higher-order corrections for identifiable reasons [46,47], *e.g.* due to flavor symmetries and constrained topologies in Feynman diagrams. An example is the process $q\bar{q} \to Zb\bar{b}$, where the leading topology is $q\bar{q} \to Zg^* \to Zb\bar{b}$, and a $t$-channel contribution favoring large $m_{bb}$ only appears when an additional jet is added to the final state. This challenges the scale variation scheme and leads to large QCD corrections [48]. Enhanced uncertainties also appear in kinematic tails of electroweak processes, for example as electroweak Sudakov logarithms and at times as large threshold corrections. Generally, for any observ-

Table 1: Scale dependence for LHC processes with only QCD particles in the final state. For each process, we report the relative scale uncertainty, the number of final state particles, and the per-particle relative scale uncertainty.

| Process | $\dfrac{\Delta\sigma}{\sigma_0}$ | | $n$ | $\dfrac{\Delta\sigma}{n\,\sigma_0}$ | |
|---|---|---|---|---|---|
| p p > j j | $+2.49 \times 10^{-1}$ | $-1.88 \times 10^{-1}$ | 2 | $+1.24 \times 10^{-1}$ | $-9.40 \times 10^{-2}$ |
| p p > b b | $+2.52 \times 10^{-1}$ | $-1.89 \times 10^{-1}$ | 2 | $+1.26 \times 10^{-1}$ | $-9.45 \times 10^{-2}$ |
| p p > t t | $+2.90 \times 10^{-1}$ | $-2.11 \times 10^{-1}$ | 2 | $+1.45 \times 10^{-1}$ | $-1.06 \times 10^{-1}$ |
| p p > j j j | $+4.38 \times 10^{-1}$ | $-2.84 \times 10^{-1}$ | 3 | $+1.46 \times 10^{-1}$ | $-9.47 \times 10^{-2}$ |
| p p > b b j | $+4.41 \times 10^{-1}$ | $-2.85 \times 10^{-1}$ | 3 | $+1.47 \times 10^{-1}$ | $-9.50 \times 10^{-2}$ |
| p p > t t j | $+4.51 \times 10^{-1}$ | $-2.90 \times 10^{-1}$ | 3 | $+1.50 \times 10^{-1}$ | $-9.67 \times 10^{-2}$ |
| p p > b b j j | $+6.18 \times 10^{-1}$ | $-3.56 \times 10^{-1}$ | 4 | $+1.54 \times 10^{-1}$ | $-8.90 \times 10^{-2}$ |
| p p > b b b b | $+6.17 \times 10^{-1}$ | $-3.56 \times 10^{-1}$ | 4 | $+1.54 \times 10^{-1}$ | $-8.90 \times 10^{-2}$ |
| p p > t t j j | $+6.14 \times 10^{-1}$ | $-3.56 \times 10^{-1}$ | 4 | $+1.53 \times 10^{-1}$ | $-8.90 \times 10^{-2}$ |
| p p > t t t t | $+6.38 \times 10^{-1}$ | $-3.65 \times 10^{-1}$ | 4 | $+1.60 \times 10^{-1}$ | $-9.12 \times 10^{-2}$ |
| p p > t t b b | $+6.21 \times 10^{-1}$ | $-3.57 \times 10^{-1}$ | 4 | $+1.55 \times 10^{-1}$ | $-8.93 \times 10^{-2}$ |
| average | | | | $+1.47 \times 10^{-1}$ | $-9.34 \times 10^{-2}$ |

able strongly sensitive to more than one relevant scale, large logarithms of their ratio tend to enhance perturbative corrections, challenging the standard estimation of their uncertainties.

All these considerations depend on the details of the process and the phase space region under consideration. While there is sufficient understanding to understand and post-dict processes for which there are large deviations at NLO not covered by varying the scale of the LO prediction, it is clear that the treatment of the uncertainty in Eq.(2) with the implementation of Eq.(7) must be generalized to cover such processes. For this purpose we will define a set of reference processes, specifically those processes populating the core of Fig. 1.

# 4 Reference-process method

To define a conservative way of estimating theory uncertainties and the corresponding nuisance parameters, we look at some of the processes which form the controlled core in Fig. 1. The processes where a scale variation by a factor two provides a quantitatively correct estimate of the size of the NLO-corrections are shown in Tab. 1, and include the QCD processes: top pair production, bottom pair production, di-jet production, and those same processes with up to two additional jets. We observe that the $K$-factors for massive and massless quarks are very similar, despite the fact that the central scale choices and the total or fiducial rates vary extensively. In addition, the relative uncertainty per final state particle only has a small variation across these processes, suggesting that the the scale uncertainty indeed simply reflects the implicit renormalization scale dependence through the corresponding power of $\alpha_s$ (as was theoretically motivated in Sec. 2). It also suggests that one could estimate the uncertainty for a new arbitrary process with $n$ particles in the final state from the approximately universal values of the re-scaled relative uncertainty $(\Delta\sigma/\sigma_0)/n$.

It suggests that a reasonable treatment for electroweak processes could be to use the set of QCD processes in Tab. 1 as a reference class by generalizing the QCD-driven uncertainty estimate from Eq.(6). In practice, one replaces their process-specific LO scale uncertainty with

$$\frac{\Delta\sigma_{\text{ref}}}{\sigma_0} = n \times \left\langle \frac{\Delta\sigma}{n\sigma_0} \right\rangle_{\text{QCD}}. \tag{13}$$

Of course, there will always be certain processes for which our simple generalization does not

describe the underlying physics and therefore will not apply. For example, $pp \rightarrow H + n$ jets production, as discussed above, is a case where the NLO corrections are enhanced by a combination of the fact that the LO process is loop-induced and has $(2 \rightarrow 1)$-kinematics, and where the rate scales like $\sigma \propto \alpha_s^{n+2}$. To accommodate such processes we modify Eq.(13) to our final proposal

$$\frac{\Delta\sigma_{\text{ref}}}{\sigma_0} = \max\left(\frac{\Delta\sigma_{\text{scale}}}{\sigma_0}, n \times \left\langle \frac{\Delta\sigma}{n\sigma_0} \right\rangle_{\text{QCD}}\right), \tag{14}$$

where the QCD reference value is computed from Eq.(9), making use of the typical behavior shown in Tab. 1. This generalization allows for a unified treatment in which the corresponding range is reliably implemented for each corresponding nuisance parameter relevant for a given (global) analysis, be it QCD or electroweak.

For our new method of computing theoretical uncertainties we again calculate the pull as

$$t_{\text{ref}} = \frac{\sigma_{\text{NLO}} - \sigma_0}{\Delta\sigma_{\text{ref}}}. \tag{15}$$

The pull distribution based on the reference process method is shown in Fig. 2, and given for specific processes in Tables 2- 4. Some processes still have large pulls, indicating significant underestimate of the uncertainty but the occurrence is much rarer and far less significant. While the reference process method is conservative in that it leads to a smaller width of the Gaussian fit in Fig. 2, it clearly does much better at the tails compared to Fig. 1 or what one would get by simply inflating the uncertainties by some arbitrary factor (a comparison is shown in Appendix A). Similar to Fig. 1, the pull is almost always greater than zero and aligns with our expectation that additional partonic channels included beyond LO tend to increase cross-section estimates. The processes with pulls $t > 2$ are dominated by reactions with photons in the final state (which suffer from the known issues of soft and collinear divergences), triple and quartic gauge boson production (with numerically significant gauge cancellations), associated $b$-pair production with flavor-locked channels and massless $b$-quarks (leading to large logarithms), and Higgs production (with large corrections to the loop-induced $(2 \rightarrow 1)$-process). For those processes the large NLO $K$-factors are understood as arising from specific physics effects, which cannot be easily extracted from lower orders in perturbation theory. For those processes, it would be interesting to investigate how our proposal fares in describing how the contribution of NNLO compares to the variation of the NLO rate.

## 5 Outlook

We have analyzed the behavior of the estimation of theory uncertainties at LO and the known NLO-QCD corrections for a large number of processes [40]. It turns out that the classic recipe of varying the renormalization and factorization scales provides a reliable uncertainty estimate for a sizeable set of QCD processes, but fails for electroweak processes, sometimes by orders of magnitude. This problem is known and related to the dominance of the implicit effects of the renormalization scale through the running of the strong coupling. We have proposed a new method of estimating theory uncertainties, based not on the scale dependence of a specific process, but on the scale dependence of the set of QCD reference processes for which the scale variation estimate is well founded and quantitatively reliable. The allowed range of cross-sections can be described by a nuisance parameter. It does not predict a nuanced distribution for that parameter, but this detail is not expected to matter if one profiles over it, as opposed to marginalizing.

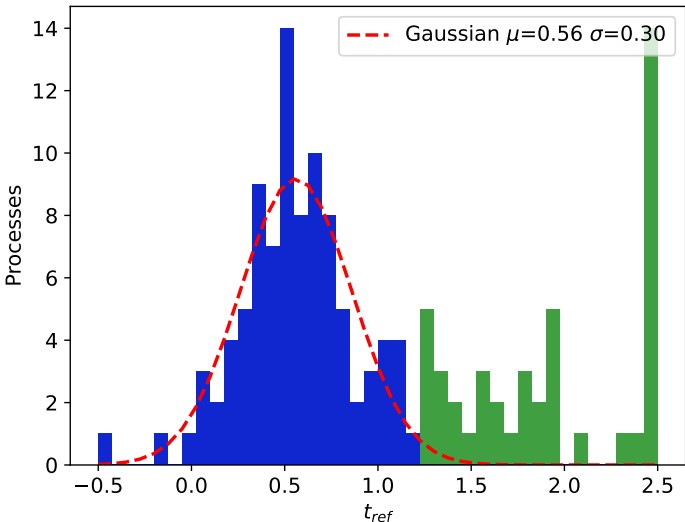

Figure 2: Performance of the uncertainty estimation in LO cross section calculations. Shown is the new pull $t_{\mathrm{ref}}$, defined in Eq.(15). Pulls greater than 2.5 are shown at 2.5. Blue entries with $|t_{\mathrm{ref}}| < 1.2$ are included in the Gaussian fit.

We illustrate the benefit of our proposed method using the original and modified pull distributions for our process dataset in Fig. 3. The pull based on our new model shows a significantly improved behavior in the tail of underestimated uncertainties which would not be expected from a naive inflation of the scale variation uncertainties (see Appendix A for a comparison). This improvement reflects a few aspects of perturbative cross-section predictions:

1. the central peak of the curve indicates that there exist universal patterns in this set of cross-sections, as observed in Tab. 1;

2. the shifted core of the pull distribution reflects, for instance, additional partonic channels;

3. the seemingly stochastic drop away from the peak indicates that for instance the central factorization and renormalization scales are indeed described as random choices of the unphysical parameters;

4. the sizable tails indicate the existence of physics effects which are not accounted for by our assumption of a homogeneous reference sample.

At this stage one might be tempted to turn the argument around and speculate what can be learned from the pull distributions about the expected size of NLO predictions. The first problem in applying Fig. 2 to completely new processes is that they might have a physics reason for large higher-order corrections. Even if that is not the case, our method is based on a sample of independent inclusive processes. Even if we assume a stochastic pattern, it does not provide predictions for individual new processes. The only way we could justify such a prediction would be through the same equivalence class, *i.e.* a physics argument based on the known properties of a given process.

While we do not advocate to use the pull distribution as a theory nuisance parameter, an estimation of theory uncertainties based on the scale variation of reference processes shows a very significant improvement over the current scheme for the reasonably inclusive processes that we have considered at a $pp$ collider with $\sqrt{s} \sim 14$ TeV. The implementation of a range of

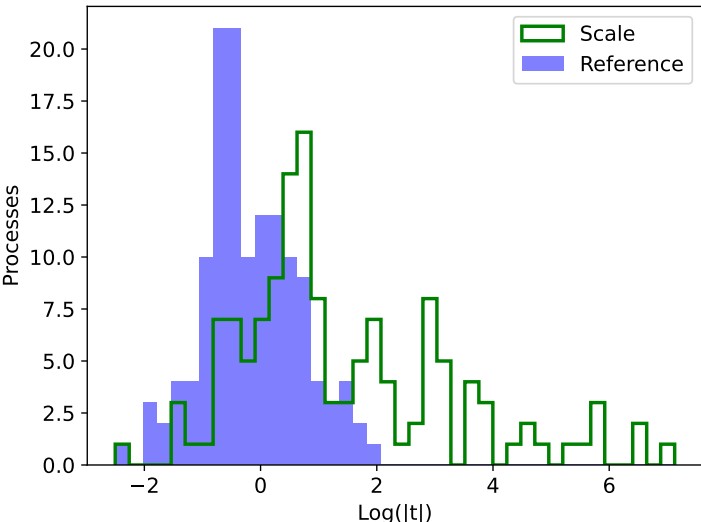

Figure 3: Comparison of the original and modified relative uncertainty via distribution of $\log t$, both for the scale-based and the reference-process definition.

predictions in terms of nuisance parameters is fairly straightforward for a profile likelihood, but would need to be better understood for Bayesian marginalization. Moreover, our reference process method is studied only for inclusive cross-sections and further studies are needed at the differential distributions in relevant observables. Differential cross-sections often contain multiple energy scales and it would be interesting to test whether the proposed method would continue to be useful for them. In addition, it would need to be tested with regard to higher orders in perturbation theory. A similar study at higher orders in perturbation theory may inform us about methods to find more such patterns.

## Acknowledgments

This work has been partly financed by the program *Internationale Spitzenforschung* of the Baden-Württemberg-Stiftung, project *Uncertainties — Teaching AI its Limits* (BWST_IF2020-010). The research of TP is supported by the Deutsche Forschungsgemeinschaft (DFG, German Research Foundation) under grant 396021762 – TRR 257 *Particle Physics Phenomenology after the Higgs Discovery*. AG/DW and BN are supported by the U.S. Department of Energy, Office of Science under grant DE-SC0009920 and contract DE-AC02-05CH11231, respectively. The work of TMPT is supported in part by the US National Science Foundation through grant PHY-2210283.

## A  Comparison to simple correction of uncertainties

The reference-process method of estimating uncertainties improves over the original scale-variation method in a significant way that cannot be matched by simple corrections of the original uncertainties. To demonstrate this, in Fig. 4 we compare the method to a simple inflation of all uncertainties by a fixed constant (while several values for the constant were studied, it is set to 3.78 in the figure, which is the mean of the ratio between the reference-process un-

certainties and the original uncertainties), and a transformation of the original uncertainties such that their mean is zero and standard deviation is one. The former fails to mitigate the tails as well as our method, and the latter distorts the core of the distribution.

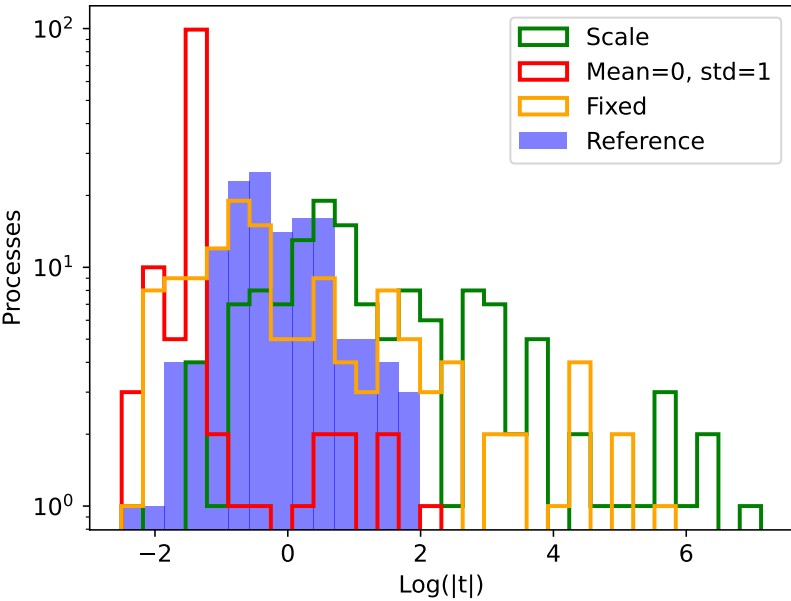

Figure 4: Comparison of the distribution of the log of the pull for original scale-variation (green) and our reference-process (blue, described in Sec. 4) methods of uncertainty estimation to two simplistic cases: a uniform inflation of the original uncertainties by a fixed constant (yellow) with value 3.78 and a scale and shift of original uncertainties such that their collective mean is zero and the standard deviation is one (red).

Table 2: A list of processes considered in Ref. [40] with their final state multiplicity ($n_{part}$), their relative uncertainty at leading order ($\Delta\sigma/\sigma_0$), and the resulting pull ($\frac{\sigma_{\mathrm{NLO}}-\sigma_0}{\Delta\sigma}$). Also included is the modified relative uncertainty at leading order ($\Delta\sigma_{\mathrm{ref}}/\sigma_0$), and the corresponding resulting modified pull ($\frac{\sigma_{\mathrm{NLO}}-\sigma_0}{\Delta\sigma_{\mathrm{ref}}}$).

| Process | $n_{\mathrm{part}}$ | $\Delta\sigma/\sigma_0$ | $\frac{\sigma_{\mathrm{NLO}}-\sigma_0}{\Delta\sigma}$ | $\Delta\sigma_{\mathrm{ref}}/\sigma_0$ | $\frac{\sigma_{\mathrm{NLO}}-\sigma_0}{\Delta\sigma_{\mathrm{ref}}}$ |
|---|---|---|---|---|---|
| p p > wpm | 1 | $1.54 \times 10^{-1}$ | 1.84 | $1.47 \times 10^{-1}$ | 1.92 |
| p p > wpm j | 2 | $1.97 \times 10^{-1}$ | 1.96 | $2.94 \times 10^{-1}$ | 1.31 |
| p p > wpm j j | 3 | $2.45 \times 10^{-1}$ | 0.59 | $4.41 \times 10^{-1}$ | 0.33 |
| p p > wpm j j j | 4 | $4.10 \times 10^{-1}$ | 0.25 | $5.88 \times 10^{-1}$ | 0.18 |
| p p > z | 1 | $1.46 \times 10^{-1}$ | 1.87 | $1.47 \times 10^{-1}$ | 1.86 |
| p p > z j | 2 | $1.93 \times 10^{-1}$ | 1.82 | $2.94 \times 10^{-1}$ | 1.19 |
| p p > z j j | 3 | $2.43 \times 10^{-1}$ | 0.56 | $4.41 \times 10^{-1}$ | 0.31 |
| p p > z j j j | 4 | $4.08 \times 10^{-1}$ | 0.27 | $5.88 \times 10^{-1}$ | 0.19 |
| p p > a j | 2 | $3.12 \times 10^{-1}$ | 5.33 | $2.94 \times 10^{-1}$ | 5.66 |
| p p > a j j | 3 | $3.28 \times 10^{-1}$ | 0.85 | $4.41 \times 10^{-1}$ | 0.63 |
| p p > w+ w- | 2 | $5.00 \times 10^{-2}$ | 7.99 | $2.94 \times 10^{-1}$ | 1.36 |
| p p > z z | 2 | $4.50 \times 10^{-2}$ | 6.46 | $2.94 \times 10^{-1}$ | 0.99 |
| p p > z wpm | 2 | $3.60 \times 10^{-2}$ | 17.09 | $2.94 \times 10^{-1}$ | 2.09 |
| p p > a a | 2 | $2.21 \times 10^{-1}$ | 7.36 | $2.94 \times 10^{-1}$ | 5.53 |
| p p > a z | 2 | $9.90 \times 10^{-2}$ | 4.73 | $2.94 \times 10^{-1}$ | 1.59 |
| p p > a wmp | 2 | $9.50 \times 10^{-2}$ | 14.88 | $2.94 \times 10^{-1}$ | 4.81 |
| p p > w+ w- j | 3 | $1.16 \times 10^{-1}$ | 2.58 | $4.41 \times 10^{-1}$ | 0.68 |
| p p > z z j | 3 | $1.09 \times 10^{-1}$ | 2.93 | $4.41 \times 10^{-1}$ | 0.73 |
| p p > z wmp j | 3 | $1.16 \times 10^{-1}$ | 2.57 | $4.41 \times 10^{-1}$ | 0.68 |
| p p > a a j | 3 | $2.03 \times 10^{-1}$ | 6.13 | $4.41 \times 10^{-1}$ | 2.83 |
| p p > a z j | 3 | $1.45 \times 10^{-1}$ | 3.23 | $4.41 \times 10^{-1}$ | 1.06 |
| p p > a wpm j | 3 | $1.37 \times 10^{-1}$ | 3.32 | $4.41 \times 10^{-1}$ | 1.03 |
| p p > w+ w+ j j | 4 | $2.54 \times 10^{-1}$ | 2.05 | $5.88 \times 10^{-1}$ | 0.89 |
| p p > w- w- j j | 4 | $2.54 \times 10^{-1}$ | 1.90 | $5.88 \times 10^{-1}$ | 0.82 |
| p p > w+ w- j j | 4 | $2.72 \times 10^{-1}$ | 0.84 | $5.88 \times 10^{-1}$ | 0.39 |
| p p > z z j j | 4 | $2.66 \times 10^{-1}$ | 1.04 | $5.88 \times 10^{-1}$ | 0.47 |
| p p > z wpm j j | 4 | $2.67 \times 10^{-1}$ | 0.51 | $5.88 \times 10^{-1}$ | 0.23 |
| p p > a a j j | 4 | $2.62 \times 10^{-1}$ | 1.50 | $5.88 \times 10^{-1}$ | 0.67 |
| p p > a z j j | 4 | $2.43 \times 10^{-1}$ | 1.24 | $5.88 \times 10^{-1}$ | 0.51 |
| p p > a wpm j j | 4 | $2.47 \times 10^{-1}$ | 0.72 | $5.88 \times 10^{-1}$ | 0.30 |
| p p > w+ w- wpm | 3 | $1.00 \times 10^{-3}$ | 610.69 | $4.41 \times 10^{-1}$ | 1.39 |
| p p > z w+ w- | 3 | $8.00 \times 10^{-3}$ | 92.39 | $4.41 \times 10^{-1}$ | 1.68 |
| p p > z z wpm | 3 | $1.00 \times 10^{-2}$ | 85.00 | $4.41 \times 10^{-1}$ | 1.93 |
| p p > z z z | 3 | $1.00 \times 10^{-3}$ | 302.75 | $4.41 \times 10^{-1}$ | 0.69 |
| p p > a w+ w- | 3 | $1.90 \times 10^{-2}$ | 42.33 | $4.41 \times 10^{-1}$ | 1.82 |
| p p > a a wpm | 3 | $4.40 \times 10^{-2}$ | 47.24 | $4.41 \times 10^{-1}$ | 4.72 |
| p p > a z wpm | 3 | $1.00 \times 10^{-3}$ | 1244.49 | $4.41 \times 10^{-1}$ | 2.82 |
| p p > a z z | 3 | $2.00 \times 10^{-2}$ | 17.24 | $4.41 \times 10^{-1}$ | 0.78 |
| p p > a a z | 3 | $5.60 \times 10^{-2}$ | 8.99 | $4.41 \times 10^{-1}$ | 1.14 |
| p p > a a a | 3 | $9.80 \times 10^{-2}$ | 17.44 | $4.41 \times 10^{-1}$ | 3.88 |
| p p > w+ w- wpm j | 4 | $1.50 \times 10^{-1}$ | 2.06 | $5.88 \times 10^{-1}$ | 0.53 |
| p p > z w+ w- j | 4 | $1.56 \times 10^{-1}$ | 1.81 | $5.88 \times 10^{-1}$ | 0.48 |
| p p > z z wmp j | 4 | $1.61 \times 10^{-1}$ | 1.88 | $5.88 \times 10^{-1}$ | 0.51 |
| p p > z z z j | 4 | $1.43 \times 10^{-1}$ | 2.21 | $5.88 \times 10^{-1}$ | 0.54 |
| p p > a a wpm j | 4 | $1.18 \times 10^{-1}$ | 3.51 | $5.88 \times 10^{-1}$ | 0.70 |
| p p > a z wpm j | 4 | $1.44 \times 10^{-1}$ | 2.30 | $5.88 \times 10^{-1}$ | 0.56 |
| p p > a z z j | 4 | $1.25 \times 10^{-1}$ | 2.96 | $5.88 \times 10^{-1}$ | 0.63 |
| p p > a a z j | 4 | $1.09 \times 10^{-1}$ | 7.57 | $5.88 \times 10^{-1}$ | 1.40 |
| p p > a a a j | 4 | $1.43 \times 10^{-1}$ | 6.72 | $5.88 \times 10^{-1}$ | 1.64 |

Table 3: A list of processes considered in Ref. [40] with their final state multiplicity ($n_{part}$), their relative uncertainty at leading order ($\Delta\sigma/\sigma_0$), and the resulting pull ($\frac{\sigma_{\mathrm{NLO}}-\sigma_0}{\Delta\sigma}$). Also included is the modified relative uncertainty at leading order ($\Delta\sigma_{\mathrm{ref}}/\sigma_0$), and the corresponding resulting modified pull ($\frac{\sigma_{\mathrm{NLO}}-\sigma_0}{\Delta\sigma_{\mathrm{ref}}}$).

| Process | $n_{\mathrm{part}}$ | $\Delta\sigma/\sigma_0$ | $\frac{\sigma_{\mathrm{NLO}}-\sigma_0}{\Delta\sigma}$ | $\Delta\sigma_{\mathrm{ref}}/\sigma_0$ | $\frac{\sigma_{\mathrm{NLO}}-\sigma_0}{\Delta\sigma_{\mathrm{ref}}}$ |
|---|---|---|---|---|---|
| p p > w+ w- w+ w- | 4 | $3.70 \times 10^{-2}$ | 20.03 | $5.88 \times 10^{-1}$ | 1.26 |
| p p > w+ w- wpm z | 4 | $4.40 \times 10^{-2}$ | 19.60 | $5.88 \times 10^{-1}$ | 1.47 |
| p p > w+ w- wpm a | 4 | $2.50 \times 10^{-2}$ | 36.35 | $5.88 \times 10^{-1}$ | 1.55 |
| p p > w+ w- z z | 4 | $4.40 \times 10^{-2}$ | 14.68 | $5.88 \times 10^{-1}$ | 1.10 |
| p p > w+ w- z a | 4 | $3.00 \times 10^{-2}$ | 25.40 | $5.88 \times 10^{-1}$ | 1.30 |
| p p > w+ w- a a | 4 | $6.00 \times 10^{-3}$ | 133.97 | $5.88 \times 10^{-1}$ | 1.37 |
| p p > wpm z z z | 4 | $5.10 \times 10^{-2}$ | 21.88 | $5.88 \times 10^{-1}$ | 1.90 |
| p p > wpm z z a | 4 | $3.60 \times 10^{-2}$ | 43.48 | $5.88 \times 10^{-1}$ | 2.66 |
| p p > wpm z a a | 4 | $1.70 \times 10^{-2}$ | 110.92 | $5.88 \times 10^{-1}$ | 3.21 |
| p p > wmp a a a | 4 | $4.00 \times 10^{-3}$ | 618.06 | $5.88 \times 10^{-1}$ | 4.21 |
| p p > z z z z | 4 | $3.80 \times 10^{-2}$ | 8.46 | $5.88 \times 10^{-1}$ | 0.55 |
| p p > z z z a | 4 | $1.90 \times 10^{-2}$ | 16.92 | $5.88 \times 10^{-1}$ | 0.55 |
| p p > z z a a | 4 | $1.00 \times 10^{-3}$ | 364.79 | $5.88 \times 10^{-1}$ | 0.62 |
| p p > z a a a | 4 | $2.30 \times 10^{-2}$ | 20.97 | $5.88 \times 10^{-1}$ | 0.82 |
| p p > a a a a | 4 | $4.70 \times 10^{-2}$ | 24.09 | $5.88 \times 10^{-1}$ | 1.93 |
| p p > j j | 2 | $2.49 \times 10^{-1}$ | 1.45 | $2.94 \times 10^{-1}$ | 1.23 |
| p p > j j j | 3 | $2.84 \times 10^{-1}$ | -0.45 | $2.80 \times 10^{-1}$ | -0.46 |
| p p > b b | 2 | $2.52 \times 10^{-1}$ | 2.86 | $2.94 \times 10^{-1}$ | 2.46 |
| p p > b b j | 3 | $4.41 \times 10^{-1}$ | 0.60 | $4.41 \times 10^{-1}$ | 0.61 |
| p p > b b j j | 4 | $6.18 \times 10^{-1}$ | 0.54 | $5.88 \times 10^{-1}$ | 0.57 |
| p p > b b b b | 4 | $6.17 \times 10^{-1}$ | 1.18 | $5.88 \times 10^{-1}$ | 1.24 |
| p p > t t | 2 | $2.90 \times 10^{-1}$ | 1.63 | $2.94 \times 10^{-1}$ | 1.61 |
| p p > t t j | 3 | $4.51 \times 10^{-1}$ | 0.68 | $4.41 \times 10^{-1}$ | 0.70 |
| p p > t t j j | 4 | $6.14 \times 10^{-1}$ | 0.53 | $5.88 \times 10^{-1}$ | 0.55 |
| p p > t t t t | 4 | $6.38 \times 10^{-1}$ | 1.63 | $5.88 \times 10^{-1}$ | 1.77 |
| p p > t t b b | 4 | $6.21 \times 10^{-1}$ | 2.20 | $5.88 \times 10^{-1}$ | 2.33 |
| p p > wpm b b | 3 | $4.23 \times 10^{-1}$ | 3.92 | $4.41 \times 10^{-1}$ | 3.76 |
| p p > z b b | 3 | $3.35 \times 10^{-1}$ | 2.31 | $4.41 \times 10^{-1}$ | 1.76 |
| p p > a b b | 3 | $5.19 \times 10^{-1}$ | 2.72 | $4.41 \times 10^{-1}$ | 3.20 |
| p p > wpm b b j | 4 | $4.25 \times 10^{-1}$ | 2.66 | $5.88 \times 10^{-1}$ | 1.92 |
| p p > z b b j | 4 | $4.24 \times 10^{-1}$ | 1.78 | $5.88 \times 10^{-1}$ | 1.29 |
| p p > a b b j | 4 | $5.12 \times 10^{-1}$ | 1.12 | $5.88 \times 10^{-1}$ | 0.98 |
| p p > t t wpm | 3 | $2.39 \times 10^{-1}$ | 2.08 | $4.41 \times 10^{-1}$ | 1.13 |
| p p > t t z | 3 | $3.05 \times 10^{-1}$ | 1.45 | $4.41 \times 10^{-1}$ | 1.00 |
| p p > t t a | 3 | $2.96 \times 10^{-1}$ | 1.52 | $4.41 \times 10^{-1}$ | 1.02 |
| p p > t t wpm j | 4 | $4.09 \times 10^{-1}$ | 1.09 | $5.88 \times 10^{-1}$ | 0.76 |
| p p > t t z j | 4 | $4.62 \times 10^{-1}$ | 0.61 | $5.88 \times 10^{-1}$ | 0.48 |
| p p > t t a j | 4 | $4.54 \times 10^{-1}$ | 0.67 | $5.88 \times 10^{-1}$ | 0.52 |
| p p > t t w+ w- | 4 | $3.09 \times 10^{-1}$ | 1.56 | $5.88 \times 10^{-1}$ | 0.82 |
| p p > t t wpm z | 4 | $2.66 \times 10^{-1}$ | 1.77 | $5.88 \times 10^{-1}$ | 0.80 |
| p p > t t wpm a | 4 | $2.54 \times 10^{-1}$ | 1.75 | $5.88 \times 10^{-1}$ | 0.76 |
| p p > t t z z | 4 | $2.93 \times 10^{-1}$ | 1.24 | $5.88 \times 10^{-1}$ | 0.62 |
| p p > t t z a | 4 | $3.01 \times 10^{-1}$ | 1.45 | $5.88 \times 10^{-1}$ | 0.74 |
| p p > t t a a | 4 | $2.84 \times 10^{-1}$ | 1.22 | $5.88 \times 10^{-1}$ | 0.59 |
| p p > tt j $$ w+ w- | 3 | $9.40 \times 10^{-2}$ | 0.28 | $4.41 \times 10^{-1}$ | 0.06 |
| p p > tt a j $$ w+ w- | 4 | $6.40 \times 10^{-2}$ | 0.38 | $5.88 \times 10^{-1}$ | 0.04 |
| p p > tt z j $$ w+ w- | 4 | $3.50 \times 10^{-2}$ | 0.08 | $5.88 \times 10^{-1}$ | 0.00 |
| p p > tt bb j $$ w+ w- | 5 | $1.38 \times 10^{-1}$ | 2.32 | $7.35 \times 10^{-1}$ | 0.44 |
| p p > tt bb j a $$ w+ w- | 6 | $1.68 \times 10^{-1}$ | 2.20 | $8.81 \times 10^{-1}$ | 0.42 |
| p p > tt bb j z $$ w+ w- | 6 | $1.87 \times 10^{-1}$ | 2.35 | $8.81 \times 10^{-1}$ | 0.50 |
| p p > w+ > t b , p p > w- > t b | 2 | $3.50 \times 10^{-2}$ | 9.57 | $2.94 \times 10^{-1}$ | 1.14 |
| p p > w+ t b a, p p > w- > t b a | 3 | $1.20 \times 10^{-2}$ | 25.73 | $4.41 \times 10^{-1}$ | 0.70 |
| p p > w+ > t b z, p p > w- > t b z | 3 | $1.30 \times 10^{-2}$ | 33.79 | $4.41 \times 10^{-1}$ | 1.00 |

Table 4: A list of processes considered in Ref. [40] with their final state multiplicity ($n_{part}$), their relative uncertainty at leading order ($\Delta\sigma/\sigma_0$ ), and the resulting pull ($\frac{\sigma_{\text{NLO}}-\sigma_0}{\Delta\sigma}$). Also included is the modified relative uncertainty at leading order ($\Delta\sigma_{\text{ref}}/\sigma_0$), and the corresponding resulting modified pull ($\frac{\sigma_{\text{NLO}}-\sigma_0}{\Delta\sigma_{\text{ref}}}$).

| Process | $n_{\text{part}}$ | $\Delta\sigma/\sigma_0$ | $\frac{\sigma_{\text{NLO}}-\sigma_0}{\Delta\sigma}$ | $\Delta\sigma_{\text{ref}}/\sigma_0$ | $\frac{\sigma_{\text{NLO}}-\sigma_0}{\Delta\sigma_{\text{ref}}}$ |
|---|---|---|---|---|---|
| p p > h | 1 | $3.48 \times 10^{-1}$ | 3.02 | $1.47 \times 10^{-1}$ | 7.15 |
| p p > h j | 2 | $3.94 \times 10^{-1}$ | 1.77 | $2.94 \times 10^{-1}$ | 2.37 |
| p p > h j j | 3 | $5.91 \times 10^{-1}$ | 1.18 | $4.41 \times 10^{-1}$ | 1.58 |
| p p > h j j $$ w+ w- z | 3 | $2.00 \times 10^{-2}$ | -2.26 | $2.80 \times 10^{-1}$ | -0.16 |
| p p > h j j j $$ w+ w- z | 4 | $1.57 \times 10^{-1}$ | 0.61 | $5.88 \times 10^{-1}$ | 0.16 |
| p p > h wpm | 2 | $3.50 \times 10^{-2}$ | 5.24 | $2.94 \times 10^{-1}$ | 0.62 |
| p p > h wpm j | 3 | $1.07 \times 10^{-1}$ | 1.91 | $4.41 \times 10^{-1}$ | 0.46 |
| p p > h wpm j j | 4 | $2.61 \times 10^{-1}$ | 1.18 | $5.88 \times 10^{-1}$ | 0.52 |
| p p > h z | 2 | $3.50 \times 10^{-2}$ | 5.30 | $2.94 \times 10^{-1}$ | 0.63 |
| p p > h z j | 3 | $1.06 \times 10^{-1}$ | 1.86 | $4.41 \times 10^{-1}$ | 0.45 |
| p p > h z j j | 4 | $2.62 \times 10^{-1}$ | 0.78 | $5.88 \times 10^{-1}$ | 0.35 |
| p p > h w+ w- | 3 | $1.00 \times 10^{-3}$ | 284.51 | $4.41 \times 10^{-1}$ | 0.65 |
| p p > h wpm a | 3 | $7.00 \times 10^{-3}$ | 44.78 | $4.41 \times 10^{-1}$ | 0.71 |
| p p > h z wpm | 3 | $1.10 \times 10^{-2}$ | 36.99 | $4.41 \times 10^{-1}$ | 0.92 |
| p p > h z z | 3 | $1.00 \times 10^{-3}$ | 215.31 | $4.41 \times 10^{-1}$ | 0.49 |
| p p > h t t | 3 | $3.00 \times 10^{-1}$ | 0.96 | $4.41 \times 10^{-1}$ | 0.65 |
| p p > h tt j | 4 | $2.40 \times 10^{-2}$ | 11.19 | $5.88 \times 10^{-1}$ | 0.46 |
| p p > h b b | 3 | $2.81 \times 10^{-1}$ | 0.79 | $4.41 \times 10^{-1}$ | 0.51 |
| p p > h t t  j | 4 | $4.56 \times 10^{-1}$ | 0.47 | $5.88 \times 10^{-1}$ | 0.36 |
| p p > h b b  j | 4 | $4.56 \times 10^{-1}$ | 0.49 | $5.88 \times 10^{-1}$ | 0.38 |
| p p > h h | 2 | $2.95 \times 10^{-1}$ | 1.90 | $2.94 \times 10^{-1}$ | 1.90 |
| p p > h h j j $$ w+ w- z | 4 | $7.20 \times 10^{-2}$ | 0.68 | $5.88 \times 10^{-1}$ | 0.08 |
| p p > h h wpm | 3 | $9.00 \times 10^{-3}$ | 18.09 | $4.41 \times 10^{-1}$ | 0.37 |
| p p > h h wpm j | 4 | $1.42 \times 10^{-1}$ | 1.10 | $5.88 \times 10^{-1}$ | 0.27 |
| p p > h h wpm a | 4 | $3.00 \times 10^{-2}$ | 6.84 | $5.88 \times 10^{-1}$ | 0.35 |
| p p > h h z | 3 | $9.00 \times 10^{-3}$ | 17.70 | $4.41 \times 10^{-1}$ | 0.36 |
| p p > h h z j | 4 | $1.41 \times 10^{-1}$ | 1.06 | $5.88 \times 10^{-1}$ | 0.25 |
| p p > h h z a | 4 | $2.40 \times 10^{-2}$ | 5.95 | $5.88 \times 10^{-1}$ | 0.24 |
| p p > h h z z | 4 | $3.90 \times 10^{-2}$ | 4.88 | $5.88 \times 10^{-1}$ | 0.32 |
| p p > h h z wpm | 4 | $4.80 \times 10^{-2}$ | 6.68 | $5.88 \times 10^{-1}$ | 0.55 |
| p p > h h w+ w- | 4 | $3.50 \times 10^{-2}$ | 6.65 | $5.88 \times 10^{-1}$ | 0.40 |
| p p > h h t t | 4 | $3.02 \times 10^{-1}$ | 0.26 | $5.88 \times 10^{-1}$ | 0.14 |
| p p > h h tt j | 5 | $1.00 \times 10^{-3}$ | 326.09 | $7.35 \times 10^{-1}$ | 0.44 |
| p p > h h b b | 4 | $3.43 \times 10^{-1}$ | 1.10 | $5.88 \times 10^{-1}$ | 0.64 |

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
