# Peer review of "Statistical Patterns of Theory Uncertainties"

_SciPost Physics Core, doi:SciPost Phys. Core 6, 045 (2023)_

## Round 2 · Referee Report · Anonymous (Referee 1) · 2022-11-27

Strengths

1- Proposes novel way of determining theoretical uncertainties associated with scale variations, trying to identify the weakness of usual determinations and proposing to rely on some quasi-universal properties of these uncertainties to correct this weakness.
2- Interesting proposal, backed by some interesting statistical considerations from the pulls between LO and NLO predictions.
3- Article well written

Weaknesses

1- Relies on the uncerainties of a "reference" class of processes in order to estimate QCD-related theoretical uncertainties, without explaining why these should be universal
2- Justifies the choice of this new approach to determine theoretical uncertainties through a stochastic, although the distribution obtained satisfies the expectations only partially (Gaussian behaviour, but not vanishing central value and unit standard deviation).

Report

In this article, the authors reconsider the estimation of some theoretical uncertainties for LHC processes. more specitifcally, they study the estimatiion of uncertainties for cross-section predictions based on scale variations. Comparing the distribution of pull between LO and NLO predictions for a large set of processes, they observe an approximate stochastic behaviour, with a seemingly accurate determination of uncertainties for QCD processes, but an underestimation of the size of uncertainties for electroweak processes. The authors propose to improve the determination of a latter, based on the scale variation of reference processes, leading to a distribution of the pulls in better agreement with a stochastic behaviour.

The article is interesting as it proposes a novel way of determining theoretical uncertainties associated with scale variations, trying to identify the weakness of usual determinations and proposing to rely on some quasi-universal properties of these uncertainties to correct this weakness. The article is well written, and it is an interesting proposal, backed by some interesting statistical considerations from the pulls between LO and NLO predictions.

Requested changes

Before recommening this article for publication in SciPost, I would like the authors to answer two questions, if possible adding information in their current draft :

1) The authors introduce their proposal of a scale variation based on reference (QCD) processes in order to avoid an underestimation of this scale variations for processes involving EW processes where the dependence on the scale is generally insufficient to get a good grasp of theoretical uncertainties. They support this by considering the distribution of pulls which is improved with this new proposal compared to the standard case : more Gaussian, with fewer outliers, closer to stochastic expectations. However, at the beginning, the authors indicate that one would expec this distribution to have a mean of zero and a standard deviation of 1 in the stochastic case. This turns out not to be the case (the central value is 0.56 and the standard deviation is of 0.3). What should we infer from the fact that the distribution does not obey these expectations ? is the stochastic picture still appropriate even though the paramters of the distribution are not correct ?

2) Table 1 suggest some kind of universality in the uncertainties of processes involving only QCD particles, once they are dividied by the number of particles. Is this universality to be expected from theoretical arguments ? Is it just a happy numerical accident ?

  • validity: good
  • significance: good
  • originality: good
  • clarity: high
  • formatting: excellent
  • grammar: excellent

Author:  Aishik Ghosh  on 2023-02-24  [id 3400]

(in reply to Report 1 on 2022-11-27)

We thank reviewer 1 for carefully going through our manuscript and pointing out its strengths as well as areas for improvement. Our responses to the comments are given below, and changes made to the manuscript in response to the comments are referenced.

The referee writes:

1) The authors introduce their proposal of a scale variation based on reference (QCD) processes in order to avoid an underestimation of this scale variations for processes involving EW processes where the dependence on the scale is generally insufficient to get a good grasp of theoretical uncertainties. They support this by considering the distribution of pulls which is improved with this new proposal compared to the standard case : more Gaussian, with fewer outliers, closer to stochastic expectations. However, at the beginning, the authors indicate that one would expec this distribution to have a mean of zero and a standard deviation of 1 in the stochastic case. This turns out not to be the case (the central value is 0.56 and the standard deviation is of 0.3). What should we infer from the fact that the distribution does not obey these expectations ? is the stochastic picture still appropriate even though the paramters of the distribution are not correct ?

Our response:

The fact that this study reveals a Gaussian-like core in the distribution of these pulls is in itself interesting. The offset from 0 is well understood and comes from the fact that NLO cross-sections are usually larger than LO because cross-sections tend to grow as additional partonic channels are included. The final paragraph of section 4 has been updated to reflect this information: “ Similar to Fig. 1, the pull is almost always greater than zero and aligns with our expectation that additional partonic channels included beyond LO tend to increase cross-section estimates.” The reference process method mitigates the tails of the distribution in a non-trivial way but the price we pay is to have a narrower Gaussian. We do not intend to imply that the uncertainties are perfectly fixed with our method but it is a significant step in the right direction. These shortcomings are discussed in the final paragraph of section 4 the fourth item of the list in section 5.

The referee writes:

2) Table 1 suggest some kind of universality in the uncertainties of processes involving only QCD particles, once they are dividied by the number of particles. Is this universality to be expected from theoretical arguments ? Is it just a happy numerical accident ?

Our response:

We thank the referee for comment, we have now updated the text to clearly point towards the theoretical discussion. Updated text: “In addition, the relative uncertainty per final state particle only has a small variation across these processes, suggesting that the scale uncertainty indeed simply reflects the implicit renormalization scale dependence through the corresponding power of $\alpha_s$ (as was theoretically motivated in Sec. 2).”

---

## Round 2 · Referee Report · Anonymous (Referee 2) · 2022-12-13

Strengths

1- interesting idea to inspect the higher-order pull distribution on a set of processes.
2- well written.

Weaknesses

1- arbitrary procedure to attribute QCD-like uncertainties to EW processes that lacks justification.
2- only LO, no NLO uncertainties provided.
3- only inclusive cross sections, generalisation to differential quantities hard to envision.

Report

The authors inspect leading order (LO) scale uncertainties for a range of processes. While QCD processes are found to be rather well-behaved statistically, in the case of EW processes the uncertainties are often greatly underestimated compared to the NLO corrections. A new procedure is proposed that assigns uncertainties derived from reference QCD processes to EW ones to improve the compatibility.

The limitations of LO scale uncertainties for EW production is well understood: restricted topology/kinematics, new channels opening up, ...; as also noted by the authors. An extreme case is for instance the Drell-Yan process, where no $\alpha_s$ power is present at LO and thus a $\mu_R$ variation does not yield any estimate on missing higher-order corrections.
It is also well known that by including higher orders, these restrictions are gradually lifted and thus uncertainties become increasingly more reliable. In an era where NLO predictions are fully automated and readily available in many general-purpose Monte Carlo tools and even NNLO predictions are becoming available (also in public codes), I fail to see the necessity of having more robust uncertainty estimates at LO.

Given that uncertainties for EW processes are underestimated, any procedure that inflates these will naturally improve the pull distribution.
The results in Table 1 is no surprise. In fact, it can be predicted using the running of the strong coupling (here including terms up to $\beta_1$):
\[
\pm\frac{\Delta\sigma}{n \sigma_0} =
\ln(2^{\pm2}) \beta_0
\frac{\alpha_s}{2\pi}
+ \ln(2^{\pm2}) \bigl[ \beta_1 + \beta_0^2 \ln(2^{\pm2}) \tfrac{1}{2}(n+1) \bigr]
\Bigl(\frac{\alpha_s}{2\pi}\Bigr)^2
+ \mathcal{O}(\alpha_s^3)
\]
which is virtually process independent besides the central scale choice $\mu_{R,0}$ at which $\alpha_s$ is evaluated and a small $n$ dependence entering through the 2-loop running piece. For the cases $n=2,3,4$ the numerical values are (fixing $\alpha_s=0.118$):
- n=2: $\quad +1.19\times10^{-1} \quad -8.96\times10^{-2}$
- n=3: $\quad +1.24\times10^{-1} \quad -8.46\times10^{-2}$
- n=4: $\quad +1.29\times10^{-1} \quad -7.96\times10^{-2}$
which basically reproduces the content of Table 1. It is important to note that this apparent "universality" is a consequence of only considering QCD processes at the lowest order; genuine not-RGE-predictable finite terms beyond LO will spoil this picture.
This also shows that the procedure proposed in this work, which is limited to LO only, essentially is equivalent to attributing the same $\mu_R$ variation as $\alpha_s$ to the coupling $\alpha$ appearing in the EW processes. This clearly has no physical justification whatsoever and is as ad-hoc as e.g. assigning an arbitrary $\pm10\%$ uncertainty on $\alpha$ to inflate uncertainties for the EW processes.

Given the above consideration that
- only LO uncertainties are considered, which are phenomenologically of little relevance (any precision measurement where theory systematics from missing higher orders is a concern are performed at the highest available perturbative order, typically NNLO);
- the "universality" property that motivates the transfer of uncertainties from reference processes to the class of EW processes will likely not persist beyond LO, thus limiting the scope of this procedure;
- no clear path exists in dealing with differential distributions;
I struggle to see the significance of this proposed method for actual applications.

  • validity: good
  • significance: low
  • originality: ok
  • clarity: high
  • formatting: excellent
  • grammar: excellent

Author:  Aishik Ghosh  on 2023-02-24  [id 3401]

(in reply to Report 2 on 2022-12-13)

We appreciate reviewer 2 taking the time to go through our manuscript and provide useful comments. It has certainly helped improve the clarity of our manuscript and message it is trying to convey to a wide audience of experimental and theoretical particle physicists. Our responses to the comments are given below, and changes made to the manuscript in response to the comments are referenced. As a broad point, the aim of our study was to explore statistical patterns in these theory uncertainties, we found a sub-population of processes where the uncertainties are systematically underestimated and then proposed a solution. This data-driven methodology can easily be extended to find new patterns and propose new solutions when a large enough data of NNLO processes become available, however, we highlight below the importance of this study already at LO. We hope that the additional context, and comparisons provided will help clarify the significance of our work.

The referee writes:

The limitations of LO scale uncertainties for EW production is well understood: restricted topology/kinematics, new channels opening up, ...; as also noted by the authors. An extreme case is for instance the Drell-Yan process, where no $\alpha_s$ power is present at LO and thus a $\mu R$ variation does not yield any estimate on missing higher-order corrections. It is also well known that by including higher orders, these restrictions are gradually lifted and thus uncertainties become increasingly more reliable. In an era where NLO predictions are fully automated and readily available in many general-purpose Monte Carlo tools and even NNLO predictions are becoming available (also in public codes), I fail to see the necessity of having more robust uncertainty estimates at LO.

Our response:

LO simulations are widely used in experiments at the LHC, sometimes even when NLO might be available. Recent examples from ATLAS for SUSY (example: arXiv:2211.05426, list of recent ATLAS SUSY results), ATLAS for exotics (example: ATL-PHYS-PUB-2021-020, list of recent ATLAS exotics results), from CMS for SUSY (example: SUS-21-004-pas, list of recent CMS SUSY results) and CMS for exotics (example: arXiv:2212.06695, list of recent CMS exotics results). Certain standard model processes are also simulated at LO when there are multiple particles in the final state and sometimes LO simulations are preferred because NLO simulations are slow. The sentence at the end of the first paragraph in section 3 has been updated to clarify this point: “Furthermore, most searches at the LHC still use LO for generating signal samples, particularly for signal samples in supersymmetry and exotics searches and the computational cost of generating large NLO samples can be prohibitive also for other BSM searches.”

The referee writes:

Given that uncertainties for EW processes are underestimated, any procedure that inflates these will naturally improve the pull distribution.

Our response:

The reviewer brings up an important point worth cross-checking (which we had already done internally), we have now added an Appendix to the manuscript with some evidence for how our proposed procedure performs better than simply inflating the uncertainties. Please refer to Figure 4 and the discussion added in Appendix A and reproduced below: “The reference-process method of estimating uncertainties improves over the original scale-variation method in a significant way that cannot be matched by simple corrections of the original uncertainties. To demonstrate this, in Fig. 4 we compare the method to a simple inflation of all uncertainties by a fixed constant (while several values for the constant were studied, it is set to 3.78 in the figure, which is the mean of the ratio between the reference-process uncertainties and the original uncertainties), and a transformation of the original uncertainties such that their mean is zero and standard deviation is one. The former fails to mitigate the tails as well as our method, and the latter distorts the core of the distribution.”

The referee writes:

The results in Table 1 is no surprise. In fact, it can be predicted using the running of the strong coupling (here including terms up to β1): which is virtually process independent besides the central scale choice μR,0 at which αs is evaluated and a small n dependence entering through the 2-loop running piece. For the cases n=2,3,4 the numerical values are (fixing αs = 0.118): ... which basically reproduces the content of Table 1. It is important to note that this apparent "universality" is a consequence of only considering QCD processes at the lowest order; genuine not-RGE-predictable finite terms beyond LO will spoil this picture.

Our response:

It’s true that it is not surprising, and the argument put forth is already described in section 2 and we now reference this discussion at the relevant point in section 4. It’s also true that this result rests on the idea that the RGE-induced corrections are dominant, and that the intrinsic process-dependent terms are sub-leading. That in itself is also probably related to the fact that Madgraph’s default scale choice works reasonably well. Updated text now points to the theoretical discussion: “In addition, the relative uncertainty per final state particle only has a small variation across these processes, suggesting that the scale uncertainty indeed simply reflects the implicit renormalization scale dependence through the corresponding power of $\alpha_s$ (as was theoretically motivated in Sec. 2).”

The referee writes:

This also shows that the procedure proposed in this work, which is limited to LO only, essentially is equivalent to attributing the same μR variation as αs to the coupling α appearing in the EW processes. This clearly has no physical justification whatsoever and is as ad-hoc as e.g. assigning an arbitrary ±10% uncertainty on α to inflate uncertainties for the EW processes.

Our response: The comparison between our method and an arbitrary inflation of uncertainties is now added in the manuscript, as described above. We hope this helps clarify the difference.

The referee writes:

  • only LO uncertainties are considered, which are phenomenologically of little relevance (any precision measurement where theory systematics from missing higher orders is a concern are performed at the highest available perturbative order, typically NNLO);

Our response: This study is on LO because a large enough and consistent dataset of NLO and NNLO samples is not available. However, uncertainties on LO remain relevant for various experimental analyses such as the example papers referenced above. Besides that, we feel these results are interesting in their own merit, to the larger particle physics (theory + experiment) community and bring to light unexpected statistical patterns of these studied uncertainties, such as the Gaussian core of Figure 1 and Figure 2. The concluding lines in section 5 also discuss the motivation for follow up work using NNLO samples.

The referee writes:

  • the "universality" property that motivates the transfer of uncertainties from reference processes to the class of EW processes will likely not persist beyond LO, thus limiting the scope of this procedure;

Our response: This study aimed to explore patterns in theoretical uncertainties, found a sub-population of processes where the uncertainties are systematically underestimated and then proposed a better solution for those processes. When a sufficiently large and consistent dataset of NNLO processes becomes available, a similar exploration may be performed, patterns found and new physics-motivated solutions suggested to solve any newly discovered sub-populations of physics processes where the uncertainties are underestimated. We have described above why the study of LO is useful in and of itself. We hope that our work will show the community the value in such data-driven studies and spur the creation of a dataset that allows a similar comparison for NLO to NNLO in the future.

The referee writes:

  • no clear path exists in dealing with differential distributions;

Our response: The proposed reference process method could be straightforwardly extended to differential distributions by using uncertainties from simulations of the reference process, although it remains to be studied how well these uncertainties would behave. Further, BSM analyses try to be as inclusive as possible which means the total rate is already of interest, however, and one could define an analogous reference profess method for events after cuts, which would apply. Of course, the full method would have to be fleshed out in follow up work. We have modified the end of the outlook section accordingly: “Moreover, our reference process method should be further tested with regard to higher orders in perturbation theory and for differential cross sections. A similar study at higher orders in perturbation theory may inform us about methods to find more such patterns.”

The referee writes:

Given the above consideration that [above bullets] I struggle to see the significance of this proposed method for actual applications.

Our response: We hope that the additional context, and comparisons provided and responses to each bullet have helped clarify the significance of our work.

---

## Round 3 · Referee Report · Anonymous · 2023-3-6

Strengths
1- Proposes novel way of determining theoretical uncertainties associated with scale variations, trying to identify the weakness of usual determinations and proposing to rely on some quasi-universal properties of these uncertainties to correct this weakness.
2- Interesting proposal, backed by some interesting statistical considerations from the pulls between LO and NLO predictions.
3- Article well written
Report
The authors answered my questions in an appropriate manner.
Author: Aishik Ghosh on 2023-05-08 [id 3652]
(in reply to Report 3 on 2023-04-06)We thank the referee for reading through our manuscript and providing comments and we have updated it based on their feedback.
The referee writes:
Our response: This study was performed using total cross-sections and we have modified the text in the conclusion to highlight the fact that further studies are needed for differential distributions: “Moreover, our reference process method is studied only for inclusive cross-sections and further studies are needed at the differential distributions in relevant observables.”
The referee writes:
Our response: We have addressed the point about individual observables in the modifications already described above. We have also added text in the outlook section to clarify the scope of our work: “... shows a very significant improvement over the current scheme for the reasonably inclusive processes that we have considered at a pp collider with sqrt(s) ~ 14 TeV.”
The referee writes:
Our response: We have added in the section 2 a clarification of this point: “The choice of the central scale can vary depending on the physics process, it could be for example the scalar sum of transverse mass of all final state particles, the invariant mass of the system being produced, the average transverse energy of jets produced, or centre-of-mass energy of the collider.”
The referee writes:
Our response: We have updated the conclusion section to state the limitations of the current work and scope for future studies: “Differential cross-sections often contain multiple energy scales and it would be interesting to test whether the proposed method would continue to be useful for them. In addition, it would need to be tested with regard to higher orders in perturbation theory.”

---

## Round 3 · Referee Report · Anonymous · 2023-3-31

Report
The authors do not address the main criticism in the revised manuscript and I remain highly sceptical on the usefulness of the proposed method.
1. I have not doubted the use of LO predictions in experimental analyses but questioned whether the interpretation in these cases is actually limited by the robustness of the uncertainties that are assigned to those LO predictions. I have not seen convincing evidence that improving LO uncertainty estimates is a critical issue that needs to be addressed. The fact that LO predictions, which are known to only provide order-of-magnitude estimates, are used is already an indicator that these analyses do not rely crucially on this aspect of the theory predictions.
2. I have explained in detail in my initial report that
- by considering QCD processes at LO only, the "universality" property observed is simply the renormalisation group evolution.
- the proposed "reference-process method" is just a convoluted way of assigning $\mu_R$-variation uncertainties in $\alpha_s$ to the EW coupling $\alpha$, i.e. is as ad-hoc as dressing $\alpha$ with a $\pm10\%$ uncertainty.
I expected the authors to explicitly test this claim but instead they chose to consider a very naive approach of inflating all uncertainties by a constant factor. Obviously, that will perform rather poorly given that it will not take into account different powers of $\alpha$ in the varying EW component of the processes.
I therefore did this exercise for them and in the attachment to this report ("hist.pdf") a comparison is shown for the pull distribution. We simply take the number of EW bosons (W, Z, $\gamma$, ...) as a proxy of the number of $\alpha$ powers ($n_\alpha$) in the process and add in quadrature to the scale uncertainty an additional $\pm10\%$ uncertainty from $\alpha$
\[
\frac{\Delta\sigma_{\alpha_{\pm10\%}}}{\sigma_0}
\equiv
\sqrt{
\left(\frac{\Delta\sigma}{\sigma_0}\right)^2
+
(n_\alpha \cdot 0.1)^2
}
\]
There is no need for a special treatment as in Eq.(14).
The comparison shows that the two methods are virtually the same; if anything, the completely ad-hoc $\alpha$ error inflation is performing slightly better. This clearly illustrates what the "reference-process method" is effectively doing and how arbitrary it is without any physics justification.
3. The authors comment in their reply that a generalisation to differential distributions is straightforward. I strongly disagree. How do the authors envision transferring uncertainties from reference processes to e.g. observables associated with colour-neutral particles such as the $p_T$ spectrum of a Z boson? All their reference processes are pure QCD ones containing no colourless particles. Not to mention differences in fiducial cuts, ...
Author: Aishik Ghosh on 2023-04-09 [id 3566]
(in reply to Report 2 on 2023-03-31)
We thank the referee for their thoughtful comments.
We note that a very large number of experimental analyses use LO simulations, and improving and understanding the quantification of their theoretical systematic uncertainties is very important. Simply characterising the behaviour of these uncertainties in a large sample under consistent conditions is an important first step, which had not previously been studied.
Beyond that, we sketch possible directions forward to improve the uncertainties without performing NLO calculations. This is important, because while it is a common assumption in the experimental community that we must be resigned to having poorly modelled theory uncertainties, especially at LO, we show that there is hope to improve upon them. This step towards understanding the statistical patterns of theory uncertainties, finding patterns of success and failure, as we have done, is a valuable contribution to the experimental community. We show that they can sometimes be improved with a rather simple method as proposed by us in this paper, and there are certainly other methods (including the one proposed by the referee) that could also be studied. In fact, we hope that this paper sparks renewed interest and effort within the community to improve the quantification of theory uncertainties, which are the most challenging kind of uncertainties in an experimental measurement. We hope to clarify these points in our responses below.
The referee writes:
The authors do not address the main criticism in the revised manuscript and I remain highly sceptical on the usefulness of the proposed method. 1. I have not doubted the use of LO predictions in experimental analyses but questioned whether the interpretation in these cases is actually limited by the robustness of the uncertainties that are assigned to those LO predictions. I have not seen convincing evidence that improving LO uncertainty estimates is a critical issue that needs to be addressed. The fact that LO predictions, which are known to only provide order-of-magnitude estimates, are used is already an indicator that these analyses do not rely crucially on this aspect of the theory predictions.
Our response: LO scale uncertainties are used in experiments because of the lack of a better alternative. For some analyses, the LO uncertainties in BSM rates represent an important limitation on the impact of the experimental analyses on our understanding of the viable parameter space of BSM models. There is a danger of circular logic where experiments cannot use better uncertainty quantification methods because theorists do not build them and theorists do not build them because experimentalists seem to still use the existing tools available.
The referee writes:
I have explained in detail in my initial report that - by considering QCD processes at LO only, the "universality" property observed is simply the renormalisation group evolution. - the proposed "reference-process method" is just a convoluted way of assigning μR-variation uncertainties in αs to the EW coupling α, i.e. is as ad-hoc as dressing α with a ±10% uncertainty. I expected the authors to explicitly test this claim but instead they chose to consider a very naive approach of inflating all uncertainties by a constant factor. Obviously, that will perform rather poorly given that it will not take into account different powers of α in the varying EW component of the processes. I therefore did this exercise for them and in the attachment to this report ("hist.pdf") a comparison is shown for the pull distribution. We simply take the number of EW bosons (W, Z, γ, ...) as a proxy of the number of α powers (nα) in the process and add in quadrature to the scale uncertainty an additional ±10% uncertainty from α Δσα±10%σ0≡√(Δσσ0)2+(nα⋅0.1)2 There is no need for a special treatment as in Eq.(14). The comparison shows that the two methods are virtually the same; if anything, the completely ad-hoc α error inflation is performing slightly better. This clearly illustrates what the "reference-process method" is effectively doing and how arbitrary it is without any physics justification.
Our response:
The theoretical background is already discussed in Section 2. Upon careful reading of our proposed procedure it should be clear to a careful reader that it is indeed expected to perform similarly to the referee’s experiment. Thus, their results are not surprising. We do not claim that the method proposed is the ultimate solution, but a good step in the direction of improved uncertainty quantification. The discussion throughout the paper follows this tone, including the ‘Outlook’ section at the end.
The referee writes:
The authors comment in their reply that a generalisation to differential distributions is straightforward. I strongly disagree. How do the authors envision transferring uncertainties from reference processes to e.g. observables associated with colour-neutral particles such as the pT spectrum of a Z boson? All their reference processes are pure QCD ones containing no colourless particles. Not to mention differences in fiducial cuts, …
Our response: Our proposed procedure is defined in terms of an inclusive cross section as a starting point, but could have just as easily been considered as a set of bins in a differential cross section, where one can select the same binning on the reference process. Of course, differential processes often contain multiple energy scales and thus one would need to test that a reasonable description would continue to hold. This is an important point to follow up, but is beyond the scope of our work making an initial assay for inclusive rates.

---

## Round 3 · Referee Report · Anonymous · 2023-4-6

Strengths
1. An exploration of NLO/LO corrections in a large number of collider inclusive processes.
2. A potentially useful rule-of-thumb for quick estimations of such corrections in experimental analyses.
Weaknesses
1. Weak conceptual support for the proposed rule-of-thumb.
2. Overselling the generality of the approach.
3. Limitation to a specific choice of the factorization scale.
Report
I am not ready to recommend the current version of the manuscript for the SciPost publication, as in my view it does not appear to meet either of four mandatory acceptance criteria listed at https://scipost.org/SciPostPhys/about#criteria . My assessment of the significance of the manuscript largely sides with the concerns of referee #2 about the conceptual foundations and generality of the proposed approach. At the same time, the authors raise a practically relevant question about simple approximations of higher-order QCD contributions for various collider processes using the LO computations that continue to be widely used in LHC analyses. Finding such an approximation can benefit the experimental analyses in many ways, but it has been difficult to do it given the versatility of contributing QCD dynamics. I thus think that the proposed formula for estimating the theoretical uncertainty has some limited, non-zero value and could be published with the appropriate disclaimers and warnings, although not necessarily in a journal.
The key weakness of the manuscript, as I see it, is that it limits itself to exploring the "how", instead of the "why", of the observed behavior of the NLO/LO corrections. From the list of references, it is clear that the authors are aware of the large body of literature dedicated to the estimates of missing higher-order uncertainties (MHOUs) from the available lower orders and experimental measurements, for example in the Cacciari-Houdeau's approach. Not mentioned here are the articles on "improved" PDFs for LO parton showers, such as arXiv:0711.2473, 0910.4183, hep-ph/020412, which had studied in-depth the underlying issues on the example of well-understood QCD processes. The manuscript itself seems to base the estimation formula on vague definitions and a rather primitive picture of the actual QCD scattering, as e.g. reflected by the following paragraph:
"At each perturbative order, ultraviolet (UV) divergences in cross-section predictions are removed through renormalization, introducing a logarithmic dependence on an unphysical renormalization scale mu_R in the prediction. Similarly, infrared (IR) and collinear divergences are absorbed into the definition of the parton densities, introducing logarithms of an equally unphysical factorization scale mu_F . Both scales can be related through the resummation of large collinear logarithms, but generally are independent scales with different infrared and ultraviolet origins and can be chosen independently [7]."
This might pass as a sloppy description of a fixed-order calculation of a hard cross section, but it is not an adequate summary to relate radiative contributions of different orders in an arbitrary hadronic cross section. Parton distributions (not densities) do not absorb infrared and collinear divergences, while mu_R and mu_F are separate scales that are not related by resummation of collinear logarithms. The behavior of the full radiative contribution reflects the diagram topologies, color factors, flavor composition, kinematics, and leading radiation configurations. Many modern textbooks on QCD elaborate on these factors.
Furthermore, the scale HT/2 advocated in Eq. (10) is not special. Other scales are commonly used, resulting in different NLO and especially LO values, and often with as good or better description of data.
I agree with referee #2 that Eqs. (6) and (13) estimate the uncertainty due to the running of the strong coupling. This can certainly be useful, but a large part of the full radiative contribution does not have much to do with the scale dependence.
In the historic example of Drell-Yan pair production in 1970's, before the QCD theory was developed, the practitioners first realized that the fixed-target DY inclusive data at pair virtuality Q can be described by the parton-model (LO) prediction multiplied by a factor that is very close to K = 1 + 3\alpha_s(Q) in a large range of sqrt{s} and Q. The easy rule-of-thumb formula for the DY K factor has been used for many years; the manuscript follows a similar logic by proposing an empirical formula to approximate many NLO cross sections using LO cross sections. The NLO QCD computation for DY reveals the limitations of this approach. The NLO/LO K factor is so close to 1+3 *alpha_s because the fixed-target inclusive DY cross section is dominated by the _hard_ virtual correction whose color and pi factors combine to a net constant of about 3. This NLO hard correction is not related to alpha_s or PDF running. It gives a good approximation for Q and y distributions at x>0.01 (at fixed targets and the Tevatron), and it fails at the LHC or FCC-hh, where rapidly varying PDFs introduce large x-dependent terms, as well as for pT distributions dominated by soft and collinear dynamics. The 3*alpha_s term is proportional to the pi^2 term that arises in timelike processes like DY, gg->Higgs, s-channel single-top or jet production. It is absent in spacelike processes like DIS, t-channel single-top or jet production. The proposed formula does not capture the pi^2 contribution or [integrated-over] hard real-emission contributions at NLO.
There is no reason to expect that these issues will be simpler in the other processes, besides DY, or at higher orders, when new color and kinematic configurations contribute.
Requested changes
1. Rewrite the text to make it very clear that the estimation formula (13) applies to the QCD coupling dependence only for a finite list of QCD observables that the authors explored.
2. List specifically what colliders, center-of-mass energies, and QCD observables (inclusive distributions only?) can be safely described by this prescription. Note that the distinction between "QCD" and "electroweak" processes is spurious, as both QCD and EW radiation is present in the actual processes. The proposed formula makes better sense when the all-order QCD observable is dominated by t-channel Born kinematics.
3. Elaborate on the other scale choices besides HT/2.
4. The theoretical motivation for the estimation formula could be sharpened throughout the text to avoid venturing into insufficiently understood areas or overselling the prescription for the situations when it will clearly fail. With these revisions in place, the article may satisfy the acceptance criterion #4, "Provide a novel and synergetic link between different research areas."

---

## Round 3 · Author Response

List of changes
The final paragraph of section 4 has been updated to :
“ Similar to Fig. 1, the pull is almost always greater than zero and aligns with our expectation that additional partonic channels included beyond LO tend to increase cross-section estimates.”
The line in section 4 now refers back to the theoretical discussion section 2:
“In addition, the relative uncertainty per final state particle only has a small variation across these processes, suggesting that the scale uncertainty indeed simply reflects the implicit renormalization scale dependence through the corresponding power of $\alpha_s$ (as was theoretically motivated in Sec. 2).”
The sentence at the end of the first paragraph in section 3 has been updated to: “Furthermore, most searches at the LHC still use LO for generating signal samples, particularly for signal samples in supersymmetry and exotics searches and the computational cost of generating large NLO samples can be prohibitive also for other BSM searches.”
Figure 4 has been added to Appendix A with the discussion:
“The reference-process method of estimating uncertainties improves over the original scale-variation method in a significant way that cannot be matched by simple corrections of the original uncertainties. To demonstrate this, in Fig. 4 we compare the method to a simple inflation of all uncertainties by a fixed constant (while several values for the constant were studied, it is set to 3.78 in the figure, which is the mean of the ratio between the reference-process uncertainties and the original uncertainties), and a transformation of the original uncertainties such that their mean is zero and standard deviation is one. The former fails to mitigate the tails as well as our method, and the latter distorts the core of the distribution.”
The line in the last paragraph of section 5 has been updated to:
“Moreover, our reference process method should be further tested with regard to higher orders in perturbation theory and for differential cross sections. A similar study at higher orders in perturbation theory may inform us about methods to find more such patterns.”

---

## Round 3 · List of Changes

The final paragraph of section 4 has been updated to :
“ Similar to Fig. 1, the pull is almost always greater than zero and aligns with our expectation that additional partonic channels included beyond LO tend to increase cross-section estimates.”
The line in section 4 now refers back to the theoretical discussion section 2:
“In addition, the relative uncertainty per final state particle only has a small variation across these processes, suggesting that the scale uncertainty indeed simply reflects the implicit renormalization scale dependence through the corresponding power of $\alpha_s$ (as was theoretically motivated in Sec. 2).”
The sentence at the end of the first paragraph in section 3 has been updated to: “Furthermore, most searches at the LHC still use LO for generating signal samples, particularly for signal samples in supersymmetry and exotics searches and the computational cost of generating large NLO samples can be prohibitive also for other BSM searches.”
Figure 4 has been added to Appendix A with the discussion:
“The reference-process method of estimating uncertainties improves over the original scale-variation method in a significant way that cannot be matched by simple corrections of the original uncertainties. To demonstrate this, in Fig. 4 we compare the method to a simple inflation of all uncertainties by a fixed constant (while several values for the constant were studied, it is set to 3.78 in the figure, which is the mean of the ratio between the reference-process uncertainties and the original uncertainties), and a transformation of the original uncertainties such that their mean is zero and standard deviation is one. The former fails to mitigate the tails as well as our method, and the latter distorts the core of the distribution.”
The line in the last paragraph of section 5 has been updated to:
“Moreover, our reference process method should be further tested with regard to higher orders in perturbation theory and for differential cross sections. A similar study at higher orders in perturbation theory may inform us about methods to find more such patterns.”

---

## Round 4 · List of Changes

- Added context for central scale in Sec2: “The choice of the central scale can vary depending on the physics process, it could be for example the scalar sum of transverse mass of all final state particles, the invariant mass of the system being produced, the average transverse energy of jets produced, or centre-of-mass energy of the collider.”

- Updated outlook section with scope and context of our work: “... shows a very significant improvement over the current scheme for the reasonably inclusive processes that we have considered at a $pp collider with $\sqrt{s} \sim 14$~TeV.”

- Updated conclusion to highlight relevant followup studies: “Moreover, our reference process method is studied only for inclusive cross-sections and further studies are needed at the differential distributions in relevant observables.”

- Also added to conclusion more comments on scope of our work and potential follow up studies: “Differential cross-sections often contain multiple energy scales and it would be interesting to test whether the proposed method would continue to be useful for them. In addition, it would need to be tested with regard to higher orders in perturbation theory.”

---

## Editorial Decision

published